# Bilinear pooling in video-QA: empirical challenges and motivational drift from neurological parallels

Thomas Winterbottom[1], Sarah Xiao[2], Alistair McLean[3] and Noura Al Moubayed[1]

[1] Department of Computer Science, Durham University, Durham, United Kingdom
[2] Durham University Business School, Durham University, Durham, Durham, United Kingdom
[3] Carbon AI, Middlesbrough, United Kingdom

Corresponding author
Thomas Winterbottom,
thomas.i.winterbottom@durham.ac.uk

## ABSTRACT

Bilinear pooling (BLP) refers to a family of operations recently developed for fusing features from different modalities predominantly for visual question answering (VQA) models. Successive BLP techniques have yielded higher performance with lower computational expense, yet at the same time they have drifted further from the original motivational justification of bilinear models, instead becoming empirically motivated by task performance. Furthermore, despite significant success in text-image fusion in VQA, BLP has not yet gained such notoriety in video question answering (video-QA). Though BLP methods have continued to perform well on video tasks when fusing vision and non-textual features, BLP has recently been overshadowed by other vision and textual feature fusion techniques in video-QA. We aim to add a new perspective to the empirical and motivational drift in BLP. We take a step back and discuss the motivational origins of BLP, highlighting the often-overlooked parallels to neurological theories (Dual Coding Theory and The Two-Stream Model of Vision). We seek to carefully and experimentally ascertain the empirical strengths and limitations of BLP as a multimodal text-vision fusion technique in video-QA using two models (TVQA baseline and heterogeneous-memory-enchanced 'HME' model) and four datasets (TVQA, TGif-QA, MSVD-QA, and EgoVQA). We examine the impact of both simply replacing feature concatenation in the existing models with BLP, and a modified version of the TVQA baseline to accommodate BLP that we name the 'dual-stream' model. We find that our relatively simple integration of BLP does not increase, and mostly harms, performance on these video-QA benchmarks. Using our insights on recent work in BLP for video-QA results and recently proposed theoretical multimodal fusion taxonomies, we offer insight into why BLP-driven performance gain for video-QA benchmarks may be more difficult to achieve than in earlier VQA models. We share our perspective on, and suggest solutions for, the key issues we identify with BLP techniques for multimodal fusion in video-QA. We look beyond the empirical justification of BLP techniques and propose both alternatives and improvements to multimodal fusion by drawing neurological inspiration from Dual Coding Theory and the Two-Stream Model of Vision. We qualitatively highlight the potential for neurological inspirations in video-QA by identifying the relative abundance of psycholinguistically 'concrete' words in the vocabularies for each of the text components (*e.g.,* questions and answers) of the four video-QA datasets we experiment with.

Ego-VQA, MSVD-QA, TGif-QA, Dual coding theory, Two-stream model of vision

## INTRODUCTION

To solve the growing abundance of complex deep learning tasks, it is essential to
develop modelling and learning strategies with the capacity to learn complex and
nuanced multimodal relationships and representations. To this end, research efforts
in multimodal deep learning have taken aim at the relationship between vision and text
through visual question answering (VQA) (*Wu et al., 2017*; *Srivastava et al., 2020*) and
more recently video question answering (video-QA) (*Sun et al., 2021*). A particularly
notorious solution to learning multimodal relationships in VQA is the family of bilinear
pooling (BLP) operators (*Gao et al., 2016*; *Kim et al., 2017*; *Yu et al., 2017*; *Ben-younes et
al., 2017*; *Yu et al., 2018b*; *Ben-Younes et al., 2019*). A bilinear (outer product) expansion
is thought to encourage models to learn interactions between two feature spaces and has
experimentally outperformed 'simpler' vector operations (*i.e.,* concatenation and element-
wise-addition/multiplication) on VQA benchmarks. Though successive BLP techniques
focus on leveraging higher performance with lower computational expense, which we
wholeheartedly welcome, the context of their use has subtly drifted from application
in earlier bilinear models *e.g.,* where in *Lin, RoyChowdhury & Maji (2015)* the bilinear
mapping is learned between convolution maps (a tangible and visualisable parameter),
from compact BLP (*Gao et al., 2016*) onwards the bilinear mapping is learned between
indexes of deep feature vectors (a much less tangible unit of representation). Though
such changes are not necessarily problematic and the improved VQA performance they
have yielded is valuable, they represent a broader trend of the use of BLP methods in
multimodal fusion being justified only by empirical success. Furthermore, despite BLP's
history of success in text-image fusion in VQA, it has not yet gained such notoriety in
video-QA. Though BLP methods have continued to perform well on video tasks when
fusing vision and *non-textual* features (*Hu et al., 2021*; *Zhou et al., 2021*; *Pang et al., 2021*;
*Xu et al., 2021*; *Deng et al., 2021*; *Wang, Bao & Xu, 2021*; *Deb et al., 2022*; *Sudhakaran,
Escalera & Lanz, 2021*), BLP has recently been overshadowed by other vision and textual
feature fusion techniques in video-QA (*Kim et al., 2019*; *Li et al., 2019*; *Gao et al., 2019*;
*Liu et al., 2021*; *Liang et al., 2019*). In this paper, we aim to add a new perspective to the
empirical and motivational drift in BLP. Our contributions include the following: (I) We
carefully and experimentally ascertain the empirical strengths and limitations of BLP as a
multimodal text-vision fusion technique on 2 models (TVQA baseline and heterogeneous-
memory-enchanced 'HME' model) and 4 datasets (TVQA, TGif-QA, MSVD-QA and
EgoVqa). To this end, our experiments include replacing feature concatenation in the
existing models with BLP, and a modified version of the TVQA baseline to accommodate
BLP that we name the 'dual-stream' model. Furthermore, we contrast BLP (classified as a
'joint' representation by *Baltrušaitis, Ahuja & Morency (2019)*) with deep canonical cross
correlation (a 'co-ordinated representation'). We find that our relatively simple integration

of BLP does not increase, and mostly harms, performance on these video-QA benchmarks. (II) We discuss the motivational origins of BLP and share our observations of bilinearity in text-vision fusion. (III) By observing trends in recent work using BLP for multimodal video tasks and recently proposed theoretical multimodal fusion taxonomies, we offer insight into why BLP-driven performance gain for video-QA benchmarks may be more difficult to achieve than in earlier VQA models. (IV) We identify temporal alignment and inefficiency (computational resources and performance) as key issues with BLP as a multimodal text-vision fusion technique in video-QA, and highlight concatenation and attention mechanisms as an ideal alternative. (V) In parallel with the empirically justified innovations driving BLP methods, we explore the often-overlooked similarities of bilinear and multimodal fusion to neurological theories *e.g.*, Dual Coding Theory (*Paivio, 2013*; *Paivio, 2014*) and the Two-Stream Model of Vision (*Goodale & Milner, 1992*; *Milner, 2017*), and propose several potential neurologically justified alternatives and improvements to existing text-image fusion. We highlight the latent potential already in existing video-QA dataset to exploit neurological theories by presenting a qualitative analysis of occurrence of psycholinguistically 'concrete' words in the vocabularies of the textual components of the 4 video-QA we experiment with.

## BACKGROUND: BILINEAR POOLING

In this section we outline the development of BLP techniques, highlight how bilinear models parallel the two-stream model of vision, and discuss where bilinear models diverged from their original motivation.

### Concatenation

Early works use vector concatenation to project different features into a new joint feature space. *Zhou et al. (2015)* use vector concatenation on the Convolutional neural network (CNN) image and text features in their simple baseline VQA model. Similarly, *Lu et al. (2016)* concatenate image attention and textual features. Vector concatenation is a projection of both input vectors into a new 'joint' dimensional space. Vector concatenation as a multimodal feature fusion technique in VQA is considered a baseline and is generally empirically outperformed in VQA by the following bilinear techniques.

### Bilinear models

Working from the observations that "perceptual systems routinely separate 'content' from 'style'", *Tenenbaum & Freeman (2000)* proposed a bilinear framework on these two different aspects of purely visual inputs. They find that the multiplicative bilinear model provides "sufficiently expressive representations of factor interactions". The bilinear model in *Lin, RoyChowdhury & Maji (2015)* is a 'two-stream' architecture where distinct subnetworks model temporal and spatial aspects. The bilinear interactions are between the outputs of two CNN streams, resulting in a bilinear vector that is effectively an outer product directly on convolution maps (features are aggregated with sum-pooling). This makes intuitive sense as individual convolution maps represent specific patterns. It follows that learnable parameters representing the outer product between these maps

learn weightings between distinct and visualisable patterns directly. Interestingly, both (*Tenenbaum & Freeman, 2000*; *Lin, RoyChowdhury & Maji, 2015*) are reminiscent of two-stream hypothesises of visual processing in the human brain (*Goodale & Milner, 1992*; *Milner & Goodale, 2006*; *Milner & Goodale, 2008*; *Goodale, 2014*; *Milner, 2017*) (discussed in detail later). Though these models focus on only visual content, their generalisable two-factor frameworks would later be inspiration to multimodal representations. Fully bilinear representations using deep learning features can easily become impractically large, necessitating informed mathematical compromises to the bilinear expansion.

## Compact bilinear pooling

*Gao et al. (2016)* introduce Compact Bilinear Pooling, a technique combining the count sketch function (*Charikar, Chen & Farach-Colton, 2002*) and convolution theorem (*Domínguez, 2015*) in order to pool the outer product into a smaller bilinear representation. *Fukui et al. (2016)* use compact BLP in their VQA model to learn interactions between text and images *i.e.,* multimodal compact bilinear pooling (MCB). We note that for MCB, the learned outer product is no longer on convolution maps, but rather on the indexes of image and textual tensors. Intuitively, a given index of an image or textual tensor is more abstracted from visualisable meaning when compared to convolution map. As far as we are aware, no research has addressed the potential ramifications of this switch from distinct maps to feature indexes, and later usages of bilinear pooling methods continue this trend. Though MCB is significantly more efficient than full bilinear expansions, they still require relatively large latent dimension to perform well on VQA ($d \approx 16000$).

## Multimodal low-rank bilinear pooling

To further reduce the number of needed parameters, *Kim et al. (2017)* introduce multimodal low-rank bilinear pooling (MLB), which approximates the outer product weight representation $W$ by decomposing it into two rank-reduced projection matrices:

$$\vec{z} = MLB(\vec{x}, \vec{y}) = (X^T \vec{x}) \odot (Y^T \vec{y})$$
$$\vec{z} = \vec{x}^T W \vec{y} = \vec{x}^T XY^T \vec{y} = \mathbb{1}^T (X^T \vec{x} \odot Y^T \vec{y})$$

where $X \in \mathbb{R}^{m \times o}, Y \in \mathbb{R}^{n \times o}, o < min(m, n)$ is the output vector dimension, $\odot$ is element-wise multiplication of vectors or the Hadamard product, and $\mathbb{1}$ is the unity vector. MLB performs better than MCB in (*Osman & Samek, 2019*), but it is sensitive to hyperparameters and converges slowly. Furthermore, *Kim et al. (2017)* suggest using *Tanh* activation on the output of $\vec{z}$ to further increase model capacity.

## Multimodal factorised low rank bilinear pooling

*Yu et al. (2017)* propose multimodal factorised bilinear pooling (MFB) as an extension of MLB. Consider the bilinear projection matrix $\vec{W} \in \mathbb{R}^{m \times n}$ outlined above, to learn output $\vec{z} \in \mathbb{R}^o$ we need to learn $\vec{W} = [\vec{W}_0, \dots, \vec{W}_{o-1}]$. We generalise output $\vec{z}$:

$$z_i = \vec{x}^T \vec{X}_i \vec{Y}_i^T \vec{y} = \sum_{d=0}^{k-1} \vec{x}^T a_d b_d^T \vec{y} = \mathbb{1}^T (\vec{X}_i^T \vec{x} \odot \vec{Y}_i^T \vec{y}) \tag{1}$$

Note that MLB is equivalent to MFB where $k = 1$. MFB can be thought of as a two-part process: features are 'expanded' to higher-dimensional space by $\vec{W}_\sigma$ matrices, then

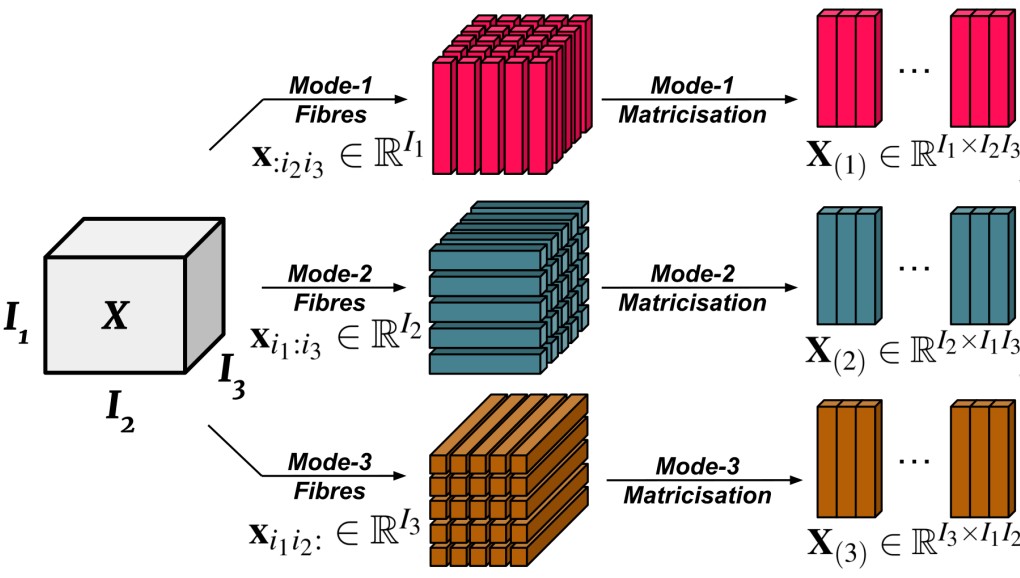

**Figure 1 Visualisation of mode-n fibres and matricisation.**

'squeezed' into a "compact ouput". The authors argue that this gives "more powerful" representational capacity in the same dimensional space than MLB.

## Multimodal tucker fusion

*Ben-younes et al. (2017)* extend the rank-reduction concept from MLB and MFB to factorise the entire bilinear tensor using tucker decomposition (*Tucker, 1966*) in their multimodal tucker fusion (MUTAN) model. We will briefly summarise the notion of rank and the mode-n product to describe the tucker decomposition model.

**Rank and mode-n product**: If $\vec{W} \in \mathbb{R}^{I_1 \times, \dots, \times I_N}$ and $\vec{V} \in \mathbb{R}^{J_n \times I_n}$ for some $n \in \{1, \dots, N\}$ then

$$\text{rank}(\vec{W} \otimes_n \vec{V}) \leq \text{rank}(\vec{W})$$

where $\otimes_n$ is the mode-n tensor product:

$$(\vec{W} \otimes_n \vec{V})(i_1, \dots, i_{n-1}, j_n, i_{n+1}, \dots, i_N) := \sum_{i_n=1}^{I_n} \vec{W}(i_1, \dots, i_{n-1}, i_n, i_{n+1}, \dots, i_N) \vec{V}(j_n, i_n)$$

In essence, the mode-n fibres (also known as mode-n vectors) of $\vec{W} \otimes_n \vec{V}$ are the mode-n fibres of $\vec{W}$ multiplied by $\vec{V}$ (proof here *Olikier, 2017*). See Fig. 1 for a visualisation of mode-n fibres. Each mode-n tensor product introduces an upper bound to the rank of the tensor. We note that conventionally, the mode-n fibres count from 1 instead of 0. We will follow this convention for the tensor product portion of our paper to avoid confusion.

The tucker decomposition of a real $3^{rd}$ order tensor $\vec{T} \in \mathbb{R}^{d_1 \times d_2 \times d_3}$ is:

$$\vec{T} = \tau \otimes_1 \vec{W}_1 \otimes_2 \vec{W}_2 \otimes_3 \vec{W}_3$$

where $\tau \in \mathbb{R}^{d_1 \times d_2 \times d_3}$ *(core tensor)*, and $\vec{W}_1, \vec{W}_2, \vec{W}_3 \in \mathbb{R}^{d_1 \times d_1}, \mathbb{R}^{d_2 \times d_2}, \mathbb{R}^{d_3 \times d_3}$ *(factor matrices)* respectively.

**MUTAN:** The MUTAN model uses a reduced rank on the core tensor to constrain representational capacity, and the factor matrices to encode full bilinear projections of the textual and visual features, and finally output an answer prediction, i.e:

$$\vec{y} = ((\tau \otimes_1 (\vec{q}^T \vec{W}_q)) \otimes_2 (\vec{v}^T \vec{W}_v)) \otimes_3 \vec{W}_o$$

where $\vec{y} \in \mathbb{R}^{|A|}$ is the answer prediction vector and $\vec{q}, \vec{v}$ are the textual and visual features respectively. A slice-wise attention mechanism is used in the MUTAN model to focus on the 'most discriminative interactions'. Multimodal tucker fusion is an empirical improvement over the preceeding BLP techniques on VQA, but it introduces complex hyperparameters to refine that are important for relatively its high performance.

## Multimodal factorised higher order bilinear pooling

All the BLP techniques discussed up to now are 'second-order', *i.e.,* take two functions as inputs. *Yu et al. (2018b)* propose multimodal factorised higher-order bilinear pooling (MFH), extending second-order BLP to 'generalised high-order pooling' by stacking multiple MFB units, *i.e.*:

$$\vec{z}^i_{exp} = MFB^i_{exp}(\vec{I}, \vec{Q}) = \vec{z}^{i-1}_{exp} \odot Dropout(\vec{U}^T \vec{I} \odot \vec{V}^T \vec{Q})$$
$$\vec{z} = SumPool(\vec{z}_{exp})$$

for $i \in \{1, \ldots, p\}$ where $\vec{I}, \vec{Q}$ are visual and text features respectively. Similar to how MFB extends MLB, MFH is MFB where $p = 1$. Though MFH slightly outperforms MFB, there has been little exploration into the theoretical benefit in generalising to higher-order BLP.

## Bilinear superdiagonal fusion

*Ben-Younes et al. (2019)* proposed another method of rank restricted bilinear pooling: Bilinear Superdiagonal Fusion (BLOCK). We will briefly outline block term decomposition before describing BLOCK.

**Block term decomposition:** Introduced in a 3-part paper (*De Lathauwer, 2008a*; *De Lathauwer, 2008b*; *De Lathauwer & Nion, 2008*), block term decomposition reformulates a bilinear matrix representation as the sum of rank restricted matrix products (contrasting low rank pooling which is represented by only a single rank restricted matrix product). By choosing the number of decompositions in the approximated sum and their rank, block-term decompositions offer greater control over the approximated bilinear model. Block term decompositions are easily extended to higher-order tensor decompositions, allowing multilinear rank restriction for multilinear models in future research. A *block term decomposition* of a tensor $\vec{W} \in \mathbb{R}^{I_1 \times, \ldots, \times I_N}$ is a decomposition of the form:

$$\vec{W} = \sum_{r=1}^R \vec{S}_r \otimes_1 \vec{U}_r^1 \otimes_2 \vec{U}_r^2 \otimes_3, \ldots, \otimes_n \vec{U}_r^n$$

where $R \in \mathbb{N}^*$ and for each $r \in \{1, \ldots, R\}, \vec{S}_r \in \mathbb{R}^{R_1 \times, \ldots, \times R_n}$ where each $\vec{S}_r$ are 'core tensors' with dimensions $R_n \leq I_n$ for $n \in \{1, \ldots, N\}$ that are used to restrict the rank of the tensor $\vec{W}.\vec{U}_r^n \in St(R_n, I_n)$ are the 'factor matrices' that intuitively expand the $n$th dimension of $\vec{S}$ back up to the original $n$th dimension of $\vec{W}$. $St(a, b)$ here refers to the Stiefel manifold, *i.e.,* $St(a, b)$: $\{\vec{Y} \in \mathbb{R}^{a \times b} : \vec{Y}^T \vec{Y} = \vec{I}_p\}$. Figure 2 visualises the block term decomposition process.

**Bilinear superdiagonal model:** The BLOCK model uses block term decompositions to learn multimodal interactions. The authors argue that since BLOCK enables "very rich (full bilinear) interactions between groups of features, while the block structure limits the

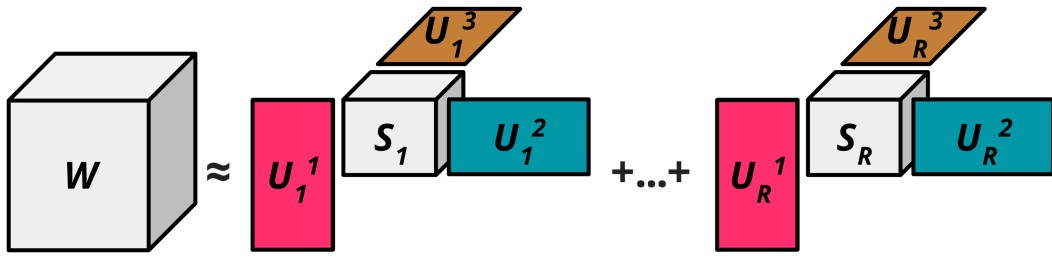

**Figure 2** Block term decomposition ($n = 3$).

complexity of the whole model'', that it is able to represent very fine grained interactions between modalities while maintaining powerful mono-modal representations. The bilinear model with inputs $\vec{x} \in \mathbb{R}^m, \vec{y} \in \mathbb{R}^n$ is projected into $o$ dimensional space with tensor products:

$$\vec{z} = \vec{W} \otimes_1 \vec{x} \otimes_2 \vec{y}$$

where $\vec{z} \in \mathbb{R}^o$. The superdiagonal BLOCK model uses a 3-dimensional block term decomposition. The decomposition of $\vec{W}$ in rank $(R_1, R_2, R_3)$ is defined as:

$$\vec{W} = \sum_{r=1}^{R} \vec{S}_r \otimes_1 \vec{U}_r^1 \otimes_2 \vec{U}_r^2 \otimes_3 \vec{U}_r^3$$

This can be written as

$$\vec{W} = \vec{S}^{bd} \otimes_1 \vec{U}^1 \otimes_2 \vec{U}^2 \otimes_3 \vec{U}^3$$

where $\vec{U}^1 = [\vec{U}_1^1, \ldots, \vec{U}_R^1]$, similarly with $\vec{U}^2$ and $\vec{U}^3$, and now $\vec{S}^{bd} \in \mathbb{R}^{RR^1 \times RR^2 \times RR^3}$. So $\vec{z}$ can now be expressed with respect to $\vec{x}$ and $\vec{y}$. Let $\hat{\vec{x}} = \vec{U}^1 \vec{x} \in \mathbb{R}^{RR^1}$ and $\hat{\vec{y}} = \vec{U}^2 \vec{y} \in \mathbb{R}^{RR^2}$. These two projections are merged by the block-superdiagonal tensor $\vec{S}^{bd}$. Each block in $\vec{S}^{bd}$ merges together blocks of size $R^1$ from $\hat{\vec{x}}$ and of size $R^2$ from $\hat{\vec{y}}$ to produce a vector of size $R^3$:

$$\vec{z}_r = \vec{S}_r \otimes_x \hat{\vec{x}}_{rR^1:(r+1)R^1} \otimes_y \hat{\vec{y}}_{rR^2:(r+1)R^2}$$

where $\hat{\vec{x}}_{i:j}$ is the vector of dimension $j - i$ containing the corresponding values of $\hat{\vec{x}}$. Finally all vectors $\vec{z}_r$ are concatenated producing $\hat{\vec{z}} \in \mathbb{R}^{RR^3}$. The final prediction vector is $\vec{z} = \vec{U}^3, \hat{\vec{z}} \in \mathbb{R}^o$. Similar to tucker fusion, the block term decomposition based fusion in BLOCK theoretically allows more nuanced control on representation size and empirically outperforms previous techniques.

## RELATED WORKS

### Bilinear pooling in video-QA with language-vision fusion

We aim to highlight and explore a broad shift away from BLP in favour of methods such as attention in video-QA benchmarks. Several video models have incorporated and contrasted BLP techniques to their own model designs for language-vision fusion tasks. *Kim et al. (2019)* find various BLP fusions perform worse than their 'dynamic modality fusion' mechanism on TVQA (*Lei et al., 2018*) and MovieQA (*Tapaswi et al., 2016*). *Li et al. (2019)* find MCB fusion performs worse on their model in ablation studies on TGIF-QA (*Jang et al., 2017*). *Chou et al. (2020)* use MLB as part of their baseline model proposed alongside their 'VQA 360°' dataset. *Gao et al. (2019)* contrast their proposed two-stream attention

mechanism to an MCB model for TGIF-QA, demonstrating a substantial performance increase over the MCB model. *Liu et al. (2021)* use MUTAN fusion between question and visual features to yield impressive results on TGif-QA, though they are outperformed by an attention based model using element-wise multiplication (*Le et al., 2020*). The Focal Visual-Text Attention network (FVTA) (*Liang et al., 2019*) is a hierarchical model that aims to dynamically select from the appropriate point across both time and modalities that outperforms an MCB approach on Movie-QA.

### Bilinear pooling in video without language-vision fusion

Where recent research in video-QA tasks (which includes textual questions as input) has moved away from BLP techniques, several video tasks that do *not* involve language have found success using BLP techniques. *Zhou et al. (2021)* use a multilevel factorised BLP based model to fuse audio and visual features for emotion recognition in videos. *Hu et al. (2021)* use compact BLP to fuse audio and 'visual long range' features for human action recognition. *Pang et al. (2021)* use MLB as part of an attention-based fusion for audio and visual features for violence detection in videos. *Xu et al. (2021)* use BLP to fuse visual features from different channels in colour image (RGB) and thermal infrared tracking (TiR) *i.e.* (RGBT). *Deng et al. (2021)* use compact BLP to fuse spatial and temporal representations of video features for action recognition. *Wang, Bao & Xu (2021)* fuse motion and appearance visual information together achieving state-of-the-art results on MSVD-QA. *Sudhakaran, Escalera & Lanz (2021)* draw design inspiration from bilinear processing of *Lin, RoyChowdhury & Maji (2015)* and MCB to propose 'Class Activation Pooling' for video action recognition. *Deb et al. (2022)* use MLB to process video features for video captioning.

## DATASETS

In this section, we outline the video-QA datasets we use in our experiments.

### MSVD-QA

*Xu et al. (2017)* argue that simply extending image-QA methods is "insufficient and suboptimal" to conduce quality video-QA, and that instead the focus should be on the temporal structure of videos. Using an natural language processing (NLP) method to automatically generate question-answer (QA) pairs from descriptions (*Heilman & Smith, 2009*), *Xu et al. (2017)* create the MSVD-QA dataset based on the Microsoft research video description corpus (*Chen & Dolan, 2011*). The dataset is made from 1970 video clips, with over 50k QA pairs in '5w' style *i.e.,* ("what", "who", "how", "when", "where").

### TGIF-QA

*Jang et al. (2017)* speculate that the relatively limited progress in video-QA compared to image-QA is "due in part to the lack of large-scale datasets with well defined tasks". As such, they introduced the TGIF-QA dataset to 'complement rather than compete' with existing VQA literature and to serve as a bridge between video-QA and video understanding. To this end, they propose 3 subsets with specific video-QA tasks that aim to take advantage of the temporal format of videos:

**Count:** Counting the number of times a specific action is repeated (*Levy & Wolf, 2015*) *e.g.*, "How many times does the girl jump?". Models output the predicted number of times the specified actions happened. (Over 30k QA pairs).

**Action:** Identify the action that is repeated a number of times in the video clip. There are over 22k multiple choice questions *e.g.*, "What does the girl do 5 times?".

**Trans:** Identifying details about a state transition (*Isola, Lim & Adelson, 2015*). There are over 58k multiple choice questions *e.g.*, "What does the man do after the goal post?".

**Frame-QA:** An image-QA split using automatically generated QA pairs from frames and captions in the TGIF dataset (*Li et al., 2016*) (over 53k multiple choice questions).

## TVQA

The TVQA dataset (*Lei et al., 2018*) is designed to address the shortcomings of previous datasets. It has significantly longer clip lengths than other datasets and is based on TV shows instead of cartoons to give it realistic video content with simple coherent narratives. It contains over 150k QA pairs. Each question is labelled with timestamps for the relevant video frames and subtitles. The questions were gathered using AMT workers. Most notably, the questions were specifically designed to encourage multimodal reasoning by asking the workers to design two-part compositional questions. The first part asks a question about a 'moment' and the second part localises the relevant moment in the video clip *i.e.,* [What/How/Where/Why/Who/…] —[when/before/after] —, *e.g.*, '*[What] was House saying [before] he leaned over the bed?*'. The authors argue this facilitates questions that require both visual and language information since "people often naturally use visual signals to ground questions in time". The authors identify certain biases in the dataset. They find that the average length of correct answers are longer than incorrect answers. They analyse the performance of their proposed baseline model with different combinations of visual and textual features on different question types they have identified. Though recent analysis has highlighted bias towards subtitles in TVQA's questions (*Winterbottom et al., 2020*), it remains an important large scale video-QA benchmark.

## EgoVQA

Most video-QA datasets focus on video-clips from the $3^{rd}$ person. *Fan (2019)* argue that 1st person video-QA has more natural use cases that real-world agents would need. As such, they propose the egocentric video-QA dataset (EgoVQA) with 609 QA pairs on 16 first-person video clips. Though the dataset is relatively small, it has a diverse set of question types (*e.g.*, 1st & 3rd person 'action' and 'who' questions, 'count', 'colour' *etc.*), and aims to generate hard and confusing incorrect answers by sampling from correct answers of the same question type. Models on EgoVQA have been shown to overfit due to its small size. To remedy this, *Fan (2019)* pretrain the baseline models on the larger YouTube2Text-QA (*Ye et al., 2017*). YouTube2Text-QA is a multiple choice dataset created from MSVD videos (*Chen & Dolan, 2011*) and questions created from YouTube2Text video description corpus (*Guadarrama et al., 2013*). YouTube2Text-QA has over 99k questions in 'what', 'who' and 'other' style.

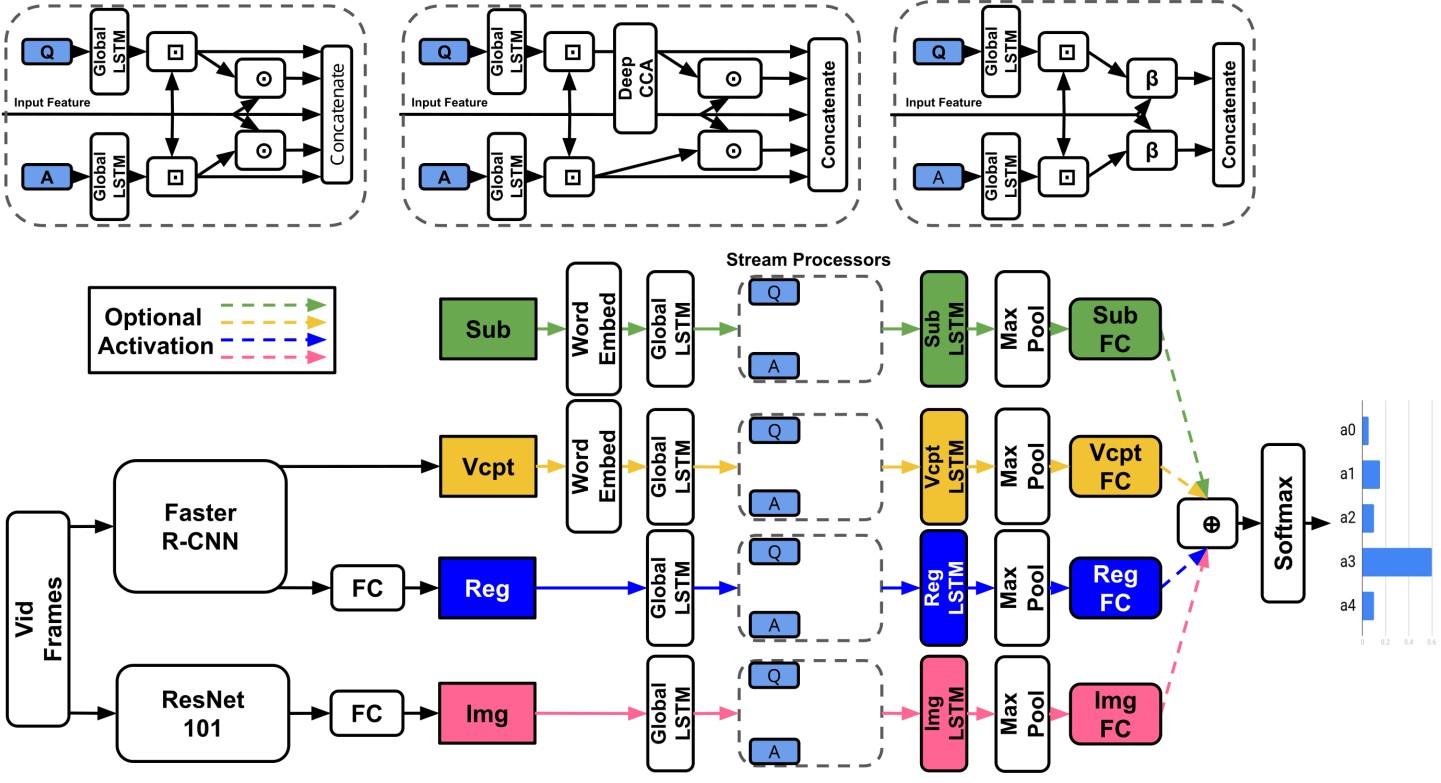

**Figure 3** TVQA Model. $\odot$/ $\oplus$ = Element-wise multiplication/addition, $\boxdot$ = context matching (*Seo et al., 2017*; *Yu et al., 2018a*), $\beta$ = BLP. Any feature streams may be enabled/disabled.

## MODELS

In this section, we describe the models used in our experiments, built from the official TVQA (https://github.com/jayleicn/TVQA) and HME-VideoQA (https://github.com/fanchenyou/HME-VideoQA) implementations.

### TVQA model

**Model Definition**: The model takes as inputs: a question $q$, five potential answers $\{a_i\}_{i=0}^{4}$, a subtitle $S$ and corresponding video-clip $V$, and outputs the predicted answer. As the model can either use the entire video-clip and subtitle or only the parts specified in the timestamp, we refer to the sections of video and subtitle used as segments from now on. Figure 3 demonstrates the textual and visual streams and their associated features in model architecture.

**ImageNet features:** Each frame is processed by a ResNet101 (*He et al., 2016*) pretrained on ImageNet (*Deng et al., 2009*) to produce a 2048-d vector. These vectors are then L2-normalised and stacked in frame order: $V^{img} \in \mathbb{R}^{f \times 2048}$ where $f$ is the number of frames used in the video segment.

**Regional features:** Each frame is processed by a Faster R-CNN (*Ren et al., 2015*) trained on Visual Genome (*Krishna et al., 2017*) in order to detect objects. Each detected object in

the frame is given a bounding box, and has an affiliated 2048-d feature extracted. Since there are multiple objects detected per frame (we cap it at 20 per frame), it is difficult to efficiently represent this in time sequences (*Lei et al., 2018*). The model uses the top-K regions for all detected labels in the segment as in (*Anderson et al., 2018*) and (*Karpathy & Fei-Fei, 2015*). Hence the regional features are $V^{reg} \in \mathbb{R}^{n_{reg} \times 2048}$ where $n_{reg}$ is the number of regional features used in the segment.

**Visual concepts:** The classes or labels of the detected regional features are called 'Visual Concepts'. *Yin & Ordonez (2017)* found that simply using detected labels instead of image features gives comparable performance on image captioning tasks. Importantly they argued that combining CNN features with detected labels outperforms either approach alone. Visual concepts are represented as either GloVe (*Pennington, Socher & Manning, 2014*) or BERT (*Devlin et al., 2019*) embeddings $V^{vcpt} \in \mathbb{R}^{n_{vcpt} \times 300}$ or $\mathbb{R}^{n_{vcpt} \times 768}$ respectively, where $n_{vcpt}$ is the number of visual concepts used in the segment.

**Text features:** The model encodes the questions, answers, and subtitles using either GloVe ($\in \mathbb{R}^{300}$) or BERT embeddings ($\in \mathbb{R}^{768}$). Formally, $q \in \mathbb{R}^{n_q \times d}, \{a_i\}_{i=0}^{4} \in \mathbb{R}^{n_{a_i} \times d}, S \in \mathbb{R}^{n_s \times d}$ where $n_q, n_{a_i}, n_s$ is the number of words in $q, a_i, S$ respectively and $d = 300, 768$ for GloVe or BERT embeddings respectively.

**Context matching:** Context matching refers to context-query attention layers recently adopted in machine comprehension (*Seo et al., 2017*; *Yu et al., 2018a*). Given a context-query pair, context matching layers return 'context aware queries'.

**Model details:** Any combination of subtitles or visual features can be used. All features are mapped into word vector space through a tanh non-linear layer. They are then processed by a shared bi-directional long short-term memory (LSTM) (*Hochreiter & Schmidhuber, 1997*; *Graves & Schmidhuber, 2005*) ('Global LSTM' in Fig. 3) of output dimension 300. Features are context-matched with the question and answers. The original context vector is then concatenated with the context-aware question and answer representations and their combined element-wise product ('Stream Processor' in Fig. 3, *e.g.*, for subtitles $S$, the stream processor outputs $[F^{sub}; A^{sub,q}; A^{sub,a_{0-4}}; F^{sub} \odot A^{sub,q}; F^{sub} \odot A^{sub,a_{0-4}}] \in \mathbb{R}^{n_{sub} \times 1500}$ where $F^{sub} \in \mathbb{R}^{n_s \times 300}$. Each concatenated vector is processed by their own unique bi-directional LSTM of output dimension 600, followed by a pair of fully connected layers of output dimensions 500 and 5, both with dropout 0.5 and ReLU activation. The 5-dimensional output represents a vote for each answer. The element-wise sum of each activated feature stream is passed to a softmax producing the predicted answer index. All features remain separate through the entire network, effectively allowing the model to choose the most useful features.

## HME-VideoQA

To better handle semantic meaning through long sequential video data, recent models have integrated external 'memory' units (*Xiong, Merity & Socher, 2016*; *Sukhbaatar et al., 2015*) alongside recurrent networks to handle input features (*Gao et al., 2018*; *Zeng et al., 2017*). These external memory units are designed to encourage multiple iterations of inference between questions and video features, helping the model revise it's visual understanding as new details from the question are presented. The heterogeneous memory-enhanced

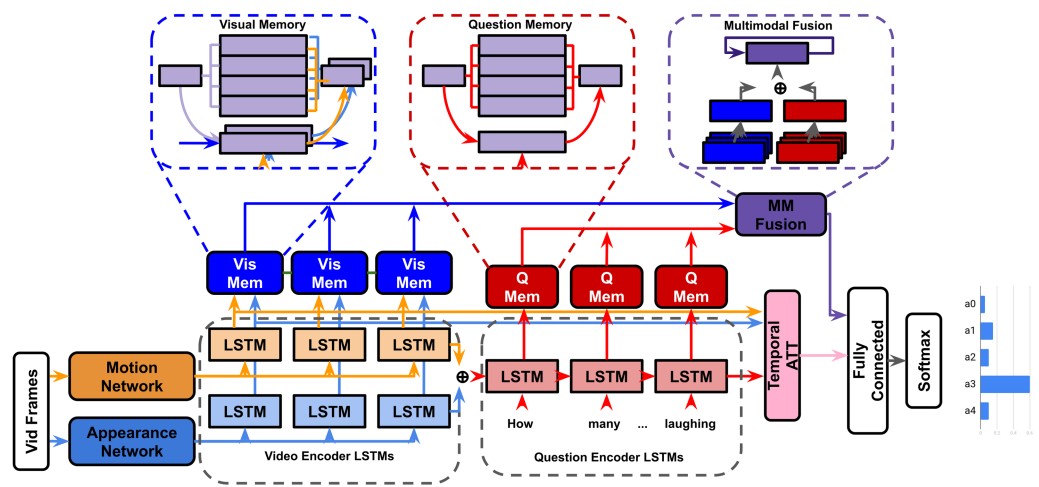

**Figure 4**    **HME model.**

video-QA model(HME) (*Fan et al., 2019*) proposes several improvements to previous memory based architectures:

**Heterogeneous read/write memory:** The memory units in HME use an attention-guided read/write mechanism to read from/update memory units respectively(the number of memory slots used is a hyperparameter). The claim is that since motion and appearance features are heterogeneous, a 'straightforward' combination of them cannot effectively describe visual information. The video memory aims to effectively fuses motion (C3D (*Tran et al., 2014*)) and appearance (ResNet (*He et al., 2016*) and VGG (*Simonyan & Zisserman, 2015*)) features by integrating them in the joint read/write operations (visual memory in Fig. 4).

**Encoder-aware question memory:** Previous memory models used a single feature vector outputted by an LSTM or gated recurrent unit (GRU) for their question representation (*Gao et al., 2018*; *Zeng et al., 2017*; *Xiong, Merity & Socher, 2016*; *Anderson et al., 2018*). HME uses an LSTM question encoder and question memory unit pair that augment each other dynamically (question memory in Fig. 4).

**Multimodal fusion unit:** The hidden states of the video and question memory units are processed by a temporal attention mechanism. The joint representation 'read' updates the fusion unit's own hidden state. The visual and question representations are ultimately fused by vector concatenation (multimodal fusion in Fig. 5). Our experiments will involve replacing this concatenation step with BLP techniques.

## EXPERIMENTS AND RESULTS

In this section we outline our experimental setup and results. We save our insights for the discussion in the next section. See our GitHub repository (https://github.com/Jumperkables/trying_blp) for both the datasets and code used in our experiments. Table 1 shows the benchmarks and state-of-the-art results for the datasets we consider in this paper.

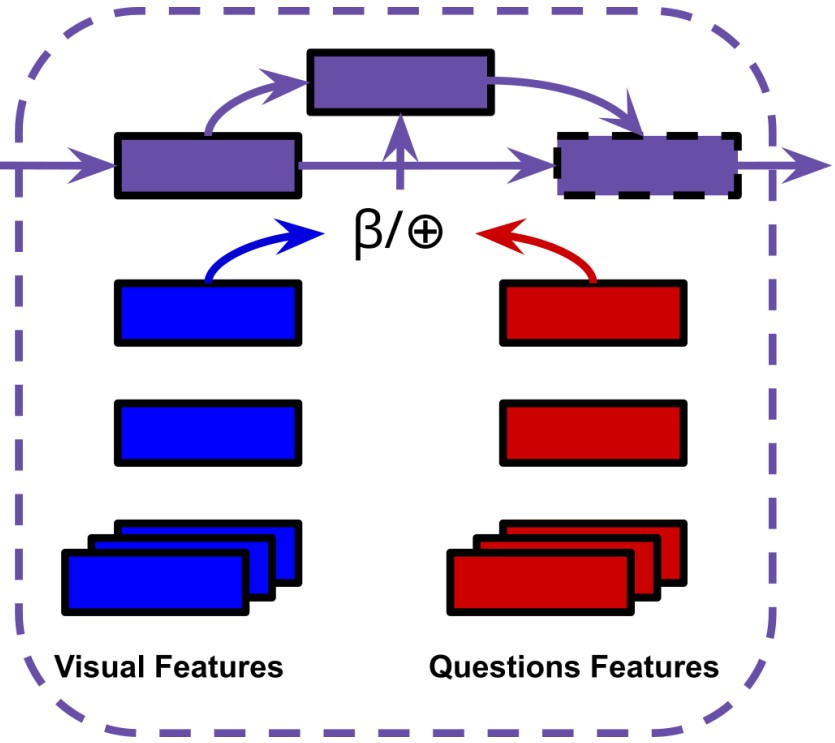

**Figure 5** ⊕ = Concatenation, β = BLP.

**Table 1** **Dataset benchmark and SoTA results to the best of our knowledge.** † = Mean L2 loss. An asterisk (*) = Results we replicated using the cited implementation.

| Dataset | Benchmark | SoTA |
|---|---|---|
| TVQA (Val) | 68.85% *Lei et al. (2018)* | 74.97% *Khan et al. (2020)* |
| TVQA (Test) | 68.48% *Lei et al. (2018)* | 72.89% *Khan et al. (2020)* |
| EgoVQA (Val 1) | 37.57% *Fan (2019)* | 45.05%* *Chenyou (2019)* |
| EgoVQA (Test 2) | 31.02% *Fan (2019)* | 43.35%* *Chenyou (2019)* |
| MSVD-QA | 32.00% *Xu et al. (2017)* | 40.30% *Guo et al. (2021)* |
| TGIF-Action | 60.77% *Jang et al. (2017)* | 84.70% *Le et al. (2020)* |
| TGIF-Count | 4.28 † *Jang et al. (2017)* | 2.19 † *Le et al. (2020)* |
| TGIF-Trans | 67.06% *Jang et al. (2017)* | 87.40% *Seo et al. (2021)* |
| TGIF-FrameQA | 49.27% *Jang et al. (2017)* | 64.80% *Le et al. (2020)* |

## Concatenation to BLP (TVQA)

As previously discussed, BLP techniques have outperformed feature concatenation on a number of VQA benchmarks. The baseline stream processor concatenates the visual feature vector with question and answer representations. Each of the five inputs to the final concatenation are 300-d. We replace the visual-question/answer concatenation with BLP (Fig. 6). All inputs to the BLP layer are 300-d, the outputs are 750-d and the hidden size is 1600 (a smaller hidden state than normal, however, the input features are also smaller compared to other uses of BLP). We make as few changes as possible to accommodate

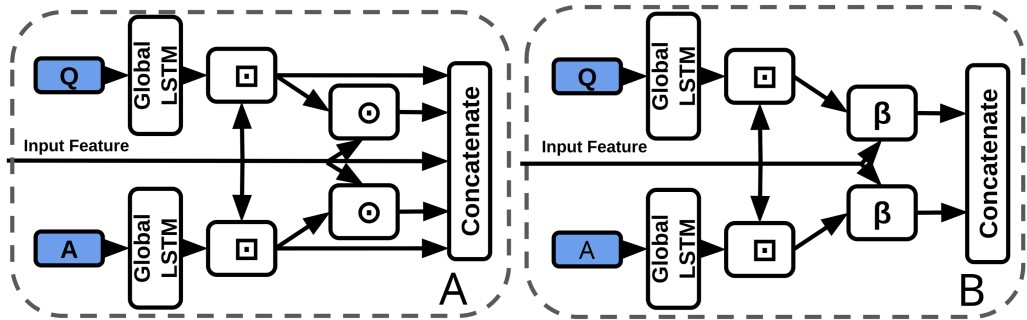

**Figure 6** **Baseline concatenation stream processor from TVQA model (left-A)** *vs* **our BLP stream processor (right-B).** $\odot$ = Element-wise multiplication, $\beta$ = BLP, $\square$ = Context Matching.

BLP, *i.e.,* we use context matching to facilitate BLP fusion by aligning visual and textual features temporally. Our experiments include models with/without subtitles or questions (Table 2).

## Dual-stream Model

We create our 'dual-stream' (Fig. 7, Table 3) model from the SI TVQA baseline model for 2 main purposes: (I) To explore the effects of a joint representation on TVQA, (II) To contrast the concatenation-replacement experiment with a model restructured specifically with BLP as a focus. The baseline BLP model keeps subtitles and other visual features completely separate up to the answer voting step. Our aim here is to create a joint representation BLP-based model similar in essence to the baseline TVQA model that fuses subtitle and visual features. As before, we use context matching to temporally align the video and text features.

## Deep CCA in TVQA

In contrast to joint representations, *Baltrušaitis, Ahuja & Morency (2019)* define 'co-ordinated representations' as a category of multimodal fusion techniques that learn "separated but co-ordinated" representations for each modality (under some constraints). *Peng, Qi & Yuan (2018)* claim that since there is often an information imbalance between modalities, learning separate modality representations can be beneficial for preserving 'exclusive and useful modality-specific characteristics'. We include one such representation, deep canonical correlation analysis (DCCA) (*Andrew et al., 2013*), in our experiments to contrast with the joint BLP models.

### CCA

Canonical cross correlation analysis (CCA) (*Hotelling, 1936*) is a method for measuring the correlations between two sets. Let $(\vec{X}_0, \vec{X}_1) \in \mathbb{R}^{d_0} \times \mathbb{R}^{d_1}$ be random vectors with covariances $(\sum_{r=00}, \sum_{r=11})$ and cross-covariance $\sum_{r=01}$. CCA finds pairs of linear projections of the two views $(w_0' \vec{X}_0, w_1' \vec{X}_1)$ that are maximally correlated:

$$\rho = (w_0^*, w_1^*) = \underset{w_0, w_1}{argmax}\ corr(w_0' \vec{X}_0, w_1' \vec{X}_1)$$

**Table 2 Concatenation replaced with BLP in the TVQA model on the TVQA Dataset.** All models use visual concepts and ImageNet features. 'No Q' indicates questions are not used as inputs i.e., answers rely purely on input features.

| Subtitles | Fusion type | Accuracy | Baseline offset |
|---|---|---|---|
| – | Concatenation | 45.94% | – |
| GloVE | Concatenation | 69.74% | – |
| BERT | Concatenation | 72.20% | – |
| – (No Q) | Concatentation | 45.58% | −0.36% |
| GloVE (No Q) | Concatentation | 68.31% | −1.42% |
| BERT (No Q) | Concatentation | 70.43% | −1.77% |
| – | MCB | **45.65%** | **−0.29%** |
| GloVE | MCB | **69.32%** | **−0.42%** |
| BERT | MCB | **71.68%** | **−0.52%** |
| – | MLB | 41.98% | −3.96% |
| GloVE | MLB | 69.30% | −0.44% |
| BERT | MLB | 69.04% | −3.16% |
| – | MFB | 41.82% | −4.12% |
| GloVE | MFB | 68.87% | −0.87% |
| BERT | MFB | 67.29% | −4.91% |
| - | MFH | 44.44% | −1.5% |
| GloVE | MFH | 68.43% | −1.31% |
| BERT | MFH | 67.29% | −4.91% |
| – | Blocktucker | 44.44% | −1.5% |
| GloVE | Blocktucker | 67.95% | −1.79% |
| BERT | Blocktucker | 67.04% | −5.16% |
| – | BLOCK | 41.09% | −4.85% |
| GloVE | BLOCK | 65.31% | −4.43% |
| BERT | BLOCK | 66.94% | −5.26% |

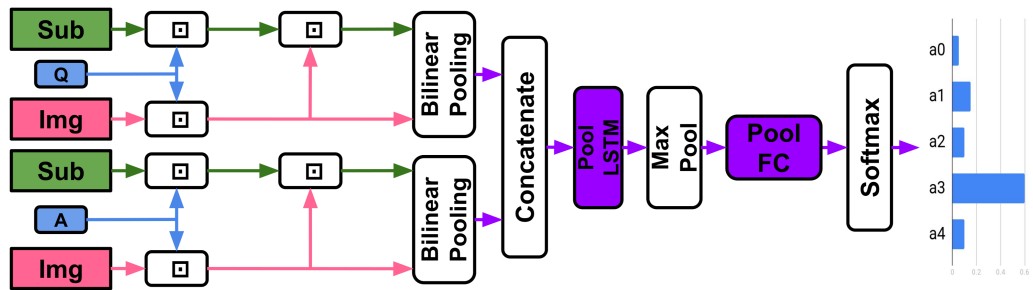

**Figure 7 Our Dual-Stream Model.** ⊡ = Context Matching.

$$= \underset{w_0, w_1}{argmax} \frac{w_0' \sum_{01} w_1}{\sqrt{w_0' \sum_{00} w_0 w_1' \sum_{11} w_1}}$$

where $\rho$ is the correlation co-efficient. As $\rho$ is invariant to the scaling of $w_0$ and $w_1$, the projections are constrained to have unit variances, and can be represented as the following maximisation:

**Table 3** Dual-Stream Results Table. 'SI' for TVQA models indicates the model is using subtitle and ImageNet feature streams only, *i.e.,* the green and pink streams in **Fig. 3**.

| Model | Text | Val Acc |
| --- | --- | --- |
| TVQA SI | GloVe | 67.78% |
| TVQA SI | BERT | 70.56% |
| Dual-Stream MCB | GloVe | 63.46% |
| Dual-Stream MCB | BERT | 60.63% |
| Dual-Stream MFH | GloVe | 62.71% |
| Dual-Stream MFH | BERT | 59.34% |

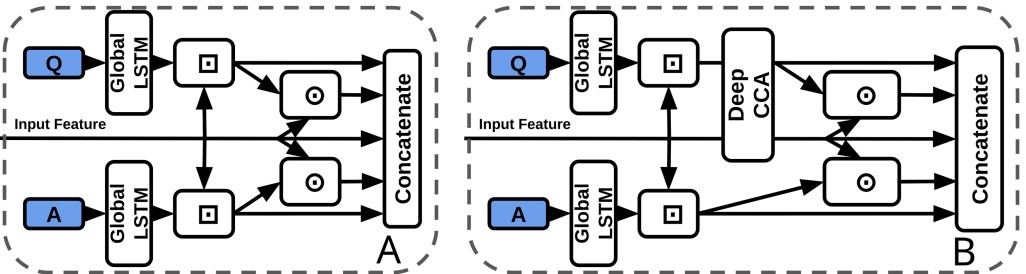

**Figure 8** Baseline concatenation stream processor from TVQA model (left-A) *vs* our DCCA stream processor (right-B). $\odot$ = Element-wise multiplication, $\boxdot$ = Context Matching.

$$argmax_{w_0, w_1} \; w_0' \sum\nolimits_{01} w_1 \; s.t \; w_0' \sum\nolimits_{00} w_0 = w_1' \sum\nolimits_{11} w_1 = \vec{1}$$

However, CCA can only model linear relationships regardless of the underlying realities in the dataset. Thus, CCA extensions were proposed, including kernel CCA (KCCA) (*Akaho, 2001*) and later DCCA.

### DCCA

DCCA is a parametric method used in multimodal neural networks that can learn non-linear transformations for input modalities. Both modalities $t, v$ are encoded in neural-network transformations $H_t, H_v = f_t(t, \theta_t), f_v(v, \theta_v)$, and then the canonical correlation between both modalities is maximised in a common subspace (*i.e.,* maximise cross-modal correlation between $H_t, H_v$).

$$\max corr(H_t, H_v) = argmax_{\theta_t, \theta_v} \; corr(f_t(t, \theta_t), f_v(v, \theta_v))$$

We use DCCA over KCCA to co-ordinate modalities in our experiments as it is generally more stable and efficient, learning more 'general' functions.

### DCCA in TVQA

We use a 2-layer DCCA module to coordinate question and context (visual or subtitle) features (Fig. 8, Table 4). Output features are the same dimensions as inputs. Though DCCA itself is not directly related to BLP, it has recently been classified as a coordinated representation (*Guo, Wang & Wang, 2019*), which contrasts a 'joint' representation.

**Table 4  DCCA in the TVQA baseline model.**

| Model | Text | Baseline Acc | DCCA Acc |
|-------|------|--------------|----------|
| VI | GloVe | 45.94% | 45.00% (−0.94%) |
| VI | BERT | – | 41.70% |
| SVI | GloVe | 69.74% | 67.91% (−1.83%) |
| SVI | BERT | 72.20% | 68.48% (−3.72%) |

**Table 5  HME-Video QA model.** The default fusion technique is concatenation. † refers to minimised L2 loss.

| Dataset | Fusion type | Val | Test |
|---------|-------------|-----|------|
| TVQA (GloVE) | Concatenation | 41.25% | N/A |
| EgoVQA-0 | Concatenation | 36.99% | 37.12% |
| EgoVQA-1 | Concatenation | 48.50% | 43.35% |
| EgoVQA-2 | Concatenation | 45.05% | 39.04% |
| MSVD-QA | Concatenation | 30.94% | 33.42% |
| TGIF-Action | Concatenation | 70.69% | 73.87% |
| TGIF-Count | Concatenation | 3.95 † | 3.92 † |
| TGIF-Trans | Concatenation | 76.33% | 78.94% |
| TGIF-FrameQA | Concatenation | 52.48% | 51.41% |
| TVQA (GloVE) | MCB | 41.09% (−0.16%) | N/A% |
| EgoVQA-0 | MCB | No Convergence | No Convergence |
| EgoVQA-1 | MCB | No Convergence | No Convergence |
| EgoVQA-2 | MCB | No Convergence | No Convergence |
| MSVD-QA | MCB | 30.85% (−0.09%) | 33.78% (+0.36%) |
| TGIF-Action | MCB | 73.56% (+2.87%) | 73.00% (−0.87%) |
| TGIF-Count | MCB | 3.95 † (+0 †) | 3.98 † (+0.06 †) |
| TGIF-Trans | MCB | 79.30% (+2.97%) | 77.10% (−1.84%) |
| TGIF-FrameQA | MCB | 51.72% (−0.76%) | 52.21% (+0.80%) |

## Concatenation to BLP (HME-VideoQA)

As described in the previous section, we replace a concatenation step in the HME model between textual and visual features with BLP (Fig. 5, corresponding to the multimodal fusion unit in Fig. 4). The goal here is to explore if BLP can better facilitate multimodal fusion in aggregated memory features (Table 5). We replicate the results from (*Fan et al., 2019*) with the HME on the MSVD, TGIF and EgoVQA datasets using the official github repository (*Chenyou, 2019*). We extract our own C3D features from the frames in the TVQA.

## DISCUSSION

### TVQA experiments

**No BLP Improvements on TVQA:** On the HME concat-to-BLP substitution model (Table 5), MCB barely changes model performance at all. We find that none of our TVQA concat-to-BLP substitutions (Table 2) yield any improvements at all, with almost

all of them performing worse overall (0.3–5%) than even the questionless concatenation model. Curiously, MCB scores the highest of all BLP techniques. The dual-stream model performs worse still, dropping accuracy by between 5–10% *vs* the baseline (Table 3). Similarly, we find that MCB performs best despite being known to require larger latent spaces to work on VQA.

**BERT impacted the most:** For the TVQA BLP-substitution models, we find the GloVe, BERT and 'no-subtitle' variations all degrade by roughly similar margins, with BERT models degrading more most often. This slight discrepancy is unsurprising as the most stable BERT baseline model is the best, and thus may degrade more on the inferior BLP variations. However, BERT's relative degradation is much more pronounced on the dual-stream models, performing 3% worse than GloVe. We theorise that here, the significant and consistent drop is potentially caused by BERT's more contextual nature no longer helping, but actively obscuring more pronounced semantic meaning learned from subtitles and questions.

**Blame smaller latent spaces?:** Naturally, bilinear representations of time series data across multiple frames or subtitles are highly GPU memory intensive. Thus we can only explore relatively small hidden dimensions (*i.e.*, 1600). However, we cannot simply conclude our poor results are due to our relatively small latent spaces because: (I) MCB is our best performing BLP technique. However, MCB has been outperformed by MFH on previous VQA models and it has been shown to require much larger latent spaces to work effectively in the first place (*Fukui et al., 2016*) (16000). (II) Our vector representations of text and images are also much smaller (300-d) compared to the larger representation dimensions conventional in previous benchmarks (*e.g.*, 2048 in *Fukui et al., 2016*). We note that $16000/2048 \approx 1600/300$, and so our latent-to-input size ratio is not substantially different to previous works.

**Unimodal biases in TVQA and joint representation:** Another explanation may come from works exploring textual biases inherent in TVQA to textual modalities (*Winterbottom et al., 2020*). BLP has been categorised as a 'joint representation'. *Baltrušaitis, Ahuja & Morency (2019)* consider representation as summarising multimodal data "in a way that exploits the complementarity and redundancy of multiple modalities". Joint representations combine unimodal signals into the same representation space. However, they struggle to handle missing data (*Baltrušaitis, Ahuja & Morency, 2019*) as they tend to preserve shared semantics while ignoring modality-specific information (*Guo, Wang & Wang, 2019*). The existence of unimodal text bias in TVQA implies BLP may perform poorly on the TVQA as a joint representation of it's features because: (I) information from either modality is consistently missing, (II) prioritising 'shared semantics' over 'modality-specific' information harms performance on TVQA. Though concatenation could also be classified as a joint representation, we argue that this observation still has merit. Theoretically, a concatenation layer can still model modality specific information(see Fig. 9), but a bilinear representation would seem to inherently entangle its inputs which would make modality specific information more challenging to learn since each parameter representing one modality is by definition weighted with the other. This may explain why our simpler BLP substitutions perform better than our more drastic 'joint' dual-stream model.

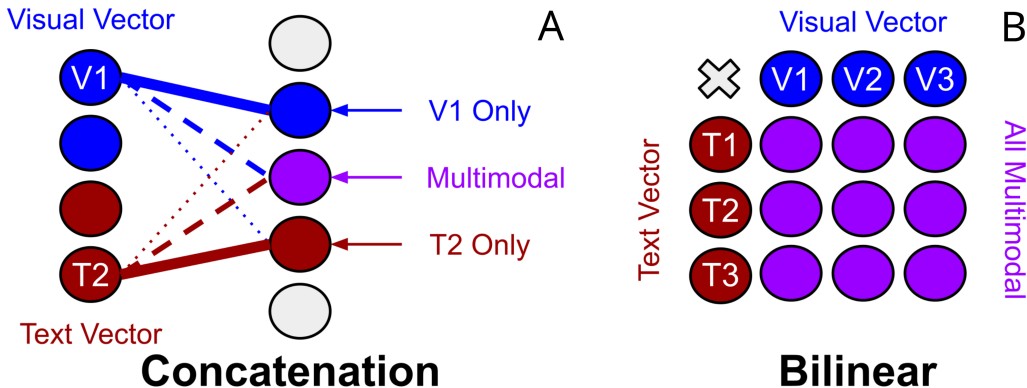

**Figure 9** **Visualisation of the differences between concatenation and bilinear representations for uni-modal processing.** Concatenation (left-A) can theoretically allow unimodal features from text or vision to process *independently* of the other modality by reducing it's weighted contribution (see 'V1 Only'). Bilinear representations (right-B) *force* multimodal interactions. It is less clear how useful 'unimodal' is processed.

**What about DCCA?:** Table 4 shows our results on the DCCA augmented TVQA models. We see a slight but noticeable performance degradation with this relatively minor alteration to the stream processor. As previously mentioned, DCCA is in some respects an opposite approach to multimodal fusion than BLP, *i.e.,* a 'coordinated representation'. The idea of a coordinated representations is to learn a separate representation for each modality , but with respect to the other. In this way, it is thought that multimodal interactions can be learned while still preserving modality-specific information that a joint representation may otherwise overlook (*Guo, Wang & Wang, 2019*; *Peng, Qi & Yuan, 2018*). DCCA specifically maximises cross-modal correlation. Without further insight from surrounding literature, it is difficult to conclude what TVQA's drop in performance using both joint and coordinated representations could mean. We will revisit this when we discuss the role of attention in multimodal fusion.

**Does context matching ruin multimodal integrity?:** The context matching technique used in the TVQA model is the birdirectional attention flow (BiDAF) module introduced in *Seo et al. (2017)*. It is used in machine comprehension between a textual context-query pair to generate query-aware context representations. BiDAF uses a 'memoryless' attention mechanism where information from each time step does not directly affect the next, which is thought to prevent early summarisation. BiDAF considers different input features at different levels of granularity. The TVQA model uses bidirectional attention flow to create context aware (visual/subtitle) question and answer representations. BiDAF can be seen as a co-ordinated representation in some regards, but it does project questions and answers representations into a new space. We use this technique to prepare our visual and question/answer features because it temporally aligns both features, giving them the same dimensional shape, conveniently allowing us to apply BLP at each time step. Since the representations generated are much more similar than the original raw features and there is some degree of information exchange, it may affect BLP's representational capacity.

Though it is worth considering these potential shortcomings, we cannot immediately assume that BiDAF would cause serious issues as earlier bilinear technique were successfully used between representations in the same modality (*Tenenbaum & Freeman, 2000*; *Gao et al., 2016*). This implies that multimodal interactions can still be learned between the more similar context-matched representations, provided the information is still present. Since BiDAF does allow visual information to be used in the TVQA baseline model, it is reasonable to assume that some of the visual information is in fact intact and exploitable for BLP. However, it is still currently unclear if context matching is fundamentally disrupting BLP and contributing to the poor results we find. We note that in BiDAF, 'memoryless' attention is implemented to avoid propagating errors through time. We argue that though this may be true and help in some circumstances, conversely, this will not allow some useful interactions to build up over time steps.

## The other datasets on HME

**BLP has no effect:** Our experiments on the EgoVQA, TGIF-QA, and MSVD-QA datasets are on concat-to-BLP substitution HME models. Our results are inconclusive. There is virtually no variation in performance between the BLP and concatenation implementations. Interestingly, EgoVQA consistently does not converge with this simple substitution. We cannot comment for certain on why this is the case. There seems to be no intuitive reason why it's 1$^{st}$ person content would cause this. Rather, we believe this is symptomatic of overfitting in training, as EgoVQA is very small and pretrained on a different dataset, and BLP techniques can sometimes have difficulties converging.

**Does better attention explain the difference?:** Attention mechanisms have been shown to improve the quality of text and visual interactions. *Yu et al. (2017)* argue that methods without attention are 'coarse joint-embedding models' which use global features that contain noisy information unhelpful in answering fine-grained questions commonly seen in VQA and video-QA. This provides strong motivation for implementing attention mechanisms alongside BLP, so that the theoretically greater representational capacity of BLP is not squandered on less useful noisy information. The TVQA model uses the previously discussed BiDAF mechanism to focus information from both modalities. However, the HME model integrates a more complex memory-based multi-hop attention mechanism. This difference may potentially highlight why the TVQA model suffers more substantially integrating BLP than the HME one.

## BLP in video-QA: problems and recommendations

We have experimented with BLP in two video-QA models and across four datasets. Our experiments show that the BLP fusion techniques popularised in VQA has not extended to increased performance to video-QA. In the preceding sections, we have supported this observation with experimental results which we contextualise by surveying the surrounding literature for BLP for multimodal video tasks. In this section, we condense our observations into a list of problems that BLP techniques pose to video-QA, and our proposal for alternatives and solutions:

**Inefficient and computationally expensive across time:** BLP as a fusion mechanism in video-QA can be exceedingly expensive due to added temporal relations. Though

propagating information from each time step through a complex text-vision multimodal fusion layer is an attractive prospect, our experiments imply that modern BLP techniques simply do not empirically perform in such a scenario. We recommend avoiding computationally expensive fusion techniques like BLP for text-image fusion *throughout* timesteps, and instead simply concatenate features at these points to save computational resources for other stages of processing (*e.g.*, attention). Furthermore, we note that any prospective fusion technique used across time will quickly encounter memory limitations that could force the hidden-size used sub-optimally low. Though summarising across time steps into condensed representations may allow more expensive BLP layers to be used on the resultant text and video representations, we instead recommend using state-of-the-art and empirically proven multimodal attention mechanisms instead (*Lei et al., 2021*; *Yang et al., 2021*). Attention mechanisms are pivotal in VQA for reducing noise and focusing on specific fine-grained details (*Yu et al., 2017*). The sheer increase in feature information when moving from still-image to video further increases the importance of attention in video-QA. Our experiments show the temporal-attention based HME model performs better when it is not degraded by BLP. Our findings are in line with that of *Long et al. (2018)* as they consider multiple different fusion methods for video classification, *i.e.*, LSTM, probability, 'feature' and attention. 'Feature' fusion is the direct connection of each modality within each local time interval, which is effectively what context matching does in the TVQA model. *Long et al. (2018)* finds temporal feature based fusion sub-par, and speculates that the burden of learning multimodal *and* temporal interactions is too heavy. Our experiments lend further evidence that for video tasks, attention-based fusion is the ideal choice.

**Problem with alignment of text and video:** As we highlight in the second subsection of our related works, BLP has yielded great performance in video tasks where it fuses the visual features with non-textual features. Audio and visual feature fusion demonstrates impressive performance on action recognition(*Hu et al., 2021*), emotion recognition (*Zhou et al., 2021*), and violence detection (*Pang et al., 2021*). Likewise, different visual representations have thrived in RGBT tracking (*Xu et al., 2021*), action recognition (*Deng et al., 2021*) and video-QA on MSVD-QA (*Wang, Bao & Xu, 2021*). On the other hand, we notice that several recent video-QA works (highlighted in the first section of our related works) have found in ablation that BLP fusion which specifically fuse visual and textual features give poor results (*Kim et al., 2019*; *Li et al., 2019*; *Gao et al., 2019*; *Liu et al., 2021*; *Liang et al., 2019*). Our observations and our experimental results highlight a pattern of poor performance for BLP in text-video fusion specifically. We demonstrate poor performance using BLP to fuse both 'BiDAF-aligned' (TVQA) and 'raw' (HME) text and video features *i.e.*, temporally aligned and unaligned respectively. As the temporally-aligned modality combinations of video-video and video-audio BLP fusion continue to succeed, we believe that the 'natural alignment' of modalities is a significant contributing factor to this performance discrepancy in video. To the best of our knowledge, we are the first to draw attention to this trend. Attention mechanisms continue to achieve state-of-the-art in video-language tasks and have been demonstrated (with visualisable attention maps) to focus on relevant video and question features. We therefore recommend using attention

mechanisms for their strong performance and relatively interpretable behaviour, and avoiding BLP for specifically video-text fusion.

**Empirically justified on VQA:** Successive BLP techniques have helped drive increased VQA performance in recent years, as such they remain an important and welcome asset to the field of multimodal machine learning. We stress that these improvements, welcome as they are, are only justified by their empirical improvements in the tasks they are applied to, and lack strong theoretical frameworks which explain their superior performance. This is entirely understandable given the infamous difficulty in interpreting how neural networks actually make decisions or exploit their training data. However, it is often claimed that such improvements are the result of some intrinsic property of the BLP operator, *e.g.*, creating 'richer multimodal representations': (*Fukui et al., 2016*) hypothesise that concatenation is not as expressive as an outer product of visual and textual features. (*Kim et al., 2017*) claim that "bilinear models provide rich representations compared with linear models". *Ben-younes et al. (2017)* claim MUTAN "focuses on modelling fine and rich interactions between image and text modalities". *Yu et al. (2018b)* claim that MFH significantly improves VQA performance "because they achieve more effective exploitation of the complex correlations between multimodal features". *Ben-Younes et al. (2019)* carefully demonstrate that the extra control over the dimensions of components in BLOCK fusion can be leveraged to achieve yet higher VQA performance, however this is attributed to it's ability "to represent very fine interactions between modalities while maintaining powerful mono-modal representations". In contrast, *Yu et al. (2017)* carefully assess and discuss the empirical improvements their MFH fusion offers on VQA. Our discussions and findings highlight the importance of being measured and nuanced when discussing the theoretical nature of multimodal fusion techniques and the benefits they bring.

# THEORETICALLY MOTIVATED OBSERVATIONS AND NEUROLOGICALLY GUIDED PROPOSALS:

BLP techniques effectively exploit mathematical innovations on bilinear expansions represented in neural networks. As previously discussed, it remains unclear why any bilinear representation would be intrinsically superior for multimodal fusion to alternatives *e.g.*, a series of non-linear fully connected layers or attention mechanisms. In this section, we share our thoughts on the properties of bilinear functions, and how they relate to neurological theories for multimodal processing in the human brain. We provide qualitative analysis of the distribution of psycholinguistic norms present in the video-QA datasets used in our experiments with which, through the lens of 'Dual Coding Theory' and the 'Two-Stream' model of vision, we propose neurologically motivated multimodal processing methodologies.

## Observations: bilinearity in BLP

**Nonlinearities in bilinear expansions:** As previously mentioned in our description of MLB, *Kim et al. (2017)* suggest using *Tanh* activation on the output of vector $\vec{z}$ to further increase model capacity. Strictly speaking, we note that adding the the non-linearity means

the representation is **no longer bilinear** as it is not linear with respect to either of its input domains. It is instead the 'same kind of non-linear' in both the input domains. We suggest that an alternative term such as 'bi-nonlinear' would more accurately describe such functions. Bilinear representations are not the most complex functions with which to learn interactions between modalities. As explored by *Yu et al. (2018b)*, we believe that higher-order interactions between features would facilitate a more realistic model of the world. The non-linear extension of bilinear or higher-order functions is a key factor to increase representational capacity.

**Outer product forces multimodal interactions:** The motivation for using bilinear methods over concatenation in VQA and video-QA was that it would enable learning more 'complex' or 'expressive' interactions between the textual and visual inputs. We note however that concatenation of inputs features should theoretically allow both a weighted multimodal combination of textual and visual units, and allow unimodal units of input features.

As visualised in Fig. 9, weights representing a bilinear expansion in a neural network each represent a multiplication of input units from each modalitiy. This appears to, in some sense, force multimodal interactions where it could possibly be advantageous to allow some degree of separation between the text and vision modalities. As discussed earlier, it is thought that 'joint' representations (*Baltrušaitis, Ahuja & Morency, 2019*) preserve shared semantics while ignoring modality-specific information (*Guo, Wang & Wang, 2019*). Though it is unclear if concatenation could effectively replicate bilinear processing while also preserving unimodal processing, it also remains unclear how exactly bilinear representations learn. For now, the successes and struggles of bilinear representations across VQA and video-QA remain justified by empirical performance on datasets.

## Proposals: neurological parallels

We have recommended that video-QA models prioritise attention mechanisms over BLP given our own experimental results and our observations of the current state-of-the-art trends. We can however still explore how bilinear models in deep learning are related to two key areas of relevant neurological research, *i.e.,* the Two-Stream model of vision (*Goodale & Milner, 1992*; *Milner, 2017*) and Dual Coding Theory (*Paivio, 2013*; *Paivio, 2014*).

**Two-stream vision:** Introduced in *Goodale & Milner (1992)*, the current consensus on primate visual processing is that it is divided into two networks or streams: The 'ventral' stream which mediates transforming the contents of visual information into 'mental furniture' that guides memory, conscious perception, and recognition; and the 'dorsal' stream which mediates the visual guidance of action. There is a wealth of evidence showing that these two subsystems are not mutually insulated from each other, but rather interconnect and contribute to one another at different stages of processing (*Milner, 2017*; *Jeannerod & Jacob, 2005*). In particular, *Jeannerod & Jacob (2005)* argue that valid comparisons between visual representation must consider the direction of fit, direction of causation and the level of conceptual content. They demonstrate that visual subsystems and behaviours inherently rely on aspects of both streams. Recently, *Milner (2017)* consider 3 potential ways these cross-stream interactions could occur: (I) Computations along

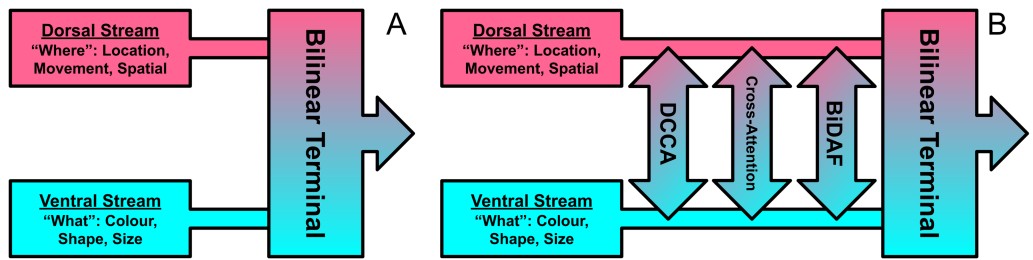

**Figure 10** Visualisation of the 1st and 3rd cross-stream scenarios for the two-stream model of vision described by *Milner (2017)*. The early bilinear model proposed by *Tenenbaum & Freeman (2000)* strikingly resembles the 1st (left-A). The 3rd and more recently favoured scenario features a continuous exchange of information across streams at multiple stages, and can be realised by introducing 'cross-talking' of deep learning features (right-B).

the two pathways are independent and combine at a 'shared terminal'(the independent processing account), (II) Processing along the separate pathways is modulated by feedback loops that transfer information from 'downstream' brain regions, including information from the complementary stream (the feedback account), (III) Information is transferred between the 2 streams at multiple stages and location along their pathways (the continuous cross-talk account).

Though *Milner (2017)* focus mostly on the 'continuous cross-talk' idea, they believe that a unifying theory would include aspects from each of these scenarios. The vision-only deep bilinear models proposed in *Tenenbaum & Freeman (2000)*; *Lin, RoyChowdhury & Maji (2015)* are strikingly reminiscent to the 1st 'shared-terminal' scenario (see Fig. 10). The bilinear framework proposed in *Tenenbaum & Freeman (2000)* focuses on splitting up 'style' and 'content', and is designed to be applied to any two-factor task. *Lin, RoyChowdhury & Maji (2015)* note but do not explore the similarities between their proposed network and the two-stream model of vision. Their bilinear CNN model aims to processes two subnetworks separately, 'what' (ventral) and 'where' (dorsal) streams, and later combine in a bilinear 'terminal'. BLP methods developed from these baselines would later focus on multimodal tasks between language and vision. As *Milner (2017)* focus mainly on their 3rd scenario (right), subsequent bilinear models that draw inspiration from the two-stream model of vision could realise the 'cross-talk' mechanism *i.e.,* using co-attention or 'co-ordinated' DCCA.

**Dual coding theory:** Dual coding theory (DCT) (*Paivio, 2013*) broadly considers the interactions between the verbal and non-verbal systems in the brain (recently surveyed in *Paivio (2014)*). DCT considers verbal and non-verbal interactions by way of 'logogens' and 'imagens' respectively, *i.e.,* units of verbal and non-verbal recognition. Imagens may be multimodal, *i.e.,* haptic, visual, smell, taste, motory etc. We should appreciate the distinction between medium and modality: image is both medium and modality and videos are an image based modality. Similarly, text is the medium through which the natural language modality is expressed. We can see parallels in multimodal deep learning and dual coding theory, with textual features as logogens and visual (or audio) features

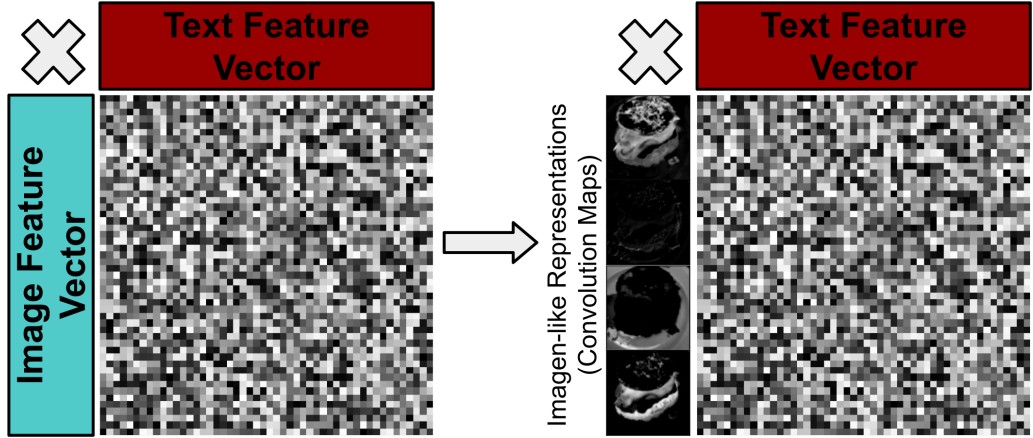

**Figure 11** Visualisation of moving from less tangible visual features to more 'imagen-like' visual features *e.g.* convolution maps of an image.

as visual(or auditory) imagens. There are many insights from DCT that could guide and drive multimodal deep learning:

(**I**) Logogens and imagens are discrete units of recognition and are often related to tangible concepts (*e.g.*, 'pictogens' (*Morton, 1979*)). By drawing inspiration from pictogen/imagen style of information representation, it could be hypothesised that multimodal models should additionally focus on deriving more tangible features (*i.e.,* discrete convolution maps previously used in vision-only bilinear models (*Lin, RoyChowdhury & Maji, 2015*)) as opposed to more abstracted 'ImageNet-style' feature vectors more commonly used in recent BLP models (see Fig. 11) are a more ideal way to represent features.

(**II**) *Bezemer & Kress (2008)* explore the differences in student's understanding when text information is presented alongside other modalities. They argue that when meaning is moved from one medium to another semiotic relations are redefined. This paradigm could be emulated to control how networks learn concepts in relation to certain modal information.

(**III**) Imagens (and potentially logogens) may be a function of many modalities, *i.e.,* one may recognise something as a function of haptic and auditory experiences alongside visual ones. We believe this implies that non-verbal modalities (vision/sound *etc.*) should be in some way grouped or aggregated, and that while DCT remains widely accepted, multimodal research should consider 'verbal *vs* non-verbal' interactions as a whole instead of focusing too intently on 'case-by-case' interactions, *i.e.,* text-vs-image and text-vs-audio. This text/non-text insight may be related to the apparent difference in text-vision video task performance previously discussed.

(**IV**) Multimodal cognitive behaviours in people can be improved by providing cues. For example, referential processing (naming an object or identifying an object from a word) has been found to additively affect free recall (recite a list of items), with the memory contribution of non-verbal codes (pictures) being twice that of verbal codes (*Paivio &*

*Lambert, 1981*). *Begg (1972)* find that free recall of 'concrete phrases' (can be visualised) of their constituent words is roughly twice that of 'abstract' phrases. However, this difference increased six-fold for concrete phrases when cued with one of the phrase words, yet using cues for abstract phrases did not help at all. This was named the 'conceptual peg' effect in DCT, and is interpreted as memory images being re-activated by 'a high imagery retrieval cue'. Given such apparent differences in human cognitive processing for 'concrete' and 'abstract' words, it may similarly be beneficial for multimodal text-vision tasks to explicitly exploit the psycholinguistic 'concreteness' word norm. Leveraging existing psycholinguistic word-norm datasets, we identify the relative abundance of concrete words in textual components of the video-QA datasets we experiment with (see Fig. 12). As the various word-norm datasets use various scoring systems for concreteness (*e.g.*, MTK40 uses a Likert scale 1-7), we rescale the scores for each dataset such that the lowest score is 0 (highly abstract), and the highest score is 1 (highly concrete). Though we cannot find a concreteness score for every word in each dataset component's vocabulary, we see that the four video-QA datasets we experiment with have more concrete than abstract words overall. Furthermore, we see that answers are on-average significantly more concrete than they are abstract, and that (as intuitively expected) visual concepts from TVQA are even more concrete.

Taking inspiration from human processing through DCT, it could be hypothesised that multimodal machine learning tasks could benefit by explicitly learning relations between 'concrete' words and their constituents, whilst treating 'abstract' words and concepts differently.

Recently proposed computational models of DCT have had many drawbacks (*Paivio, 2014*), we believe that neural networks can be a natural fit for modelling neural correlates explored in DCT and should be considered as a future modelling option.

## CONCLUSION

In light of BLP's empirical success in VQA, we have experimentally explored their use in video-QA on two models and four datasets. We find that switching from vector concatenation to BLP through simple substitution on the HME and TVQA models does not improve and in fact actively harm performance on video-QA. We find that a more substantial 'dual-stream' restructuring of the TVQA model to accommodate BLP significantly reduces performance on TVQA. Our results and observations about the downturn in successful text-vision BLP fusion in video tasks imply that naively using BLP techniques can be very detrimental in video-QA. We caution against automatically integrating bilinear pooling in video-QA models and expecting similar empirical increases as in VQA. We offer several interpretations and insights of our negative results using surrounding multimodal and neurological literature and find our results inline with trends in VQA and video-classification. To the best of our knowledge, we are the first to outline how important neurological theories *i.e.,* dual coding theory and the two-stream model of vision relate to the history of (and journey to) modern multimodal deep learning practices. We offer a few experimentally and theoretically guided suggestions to

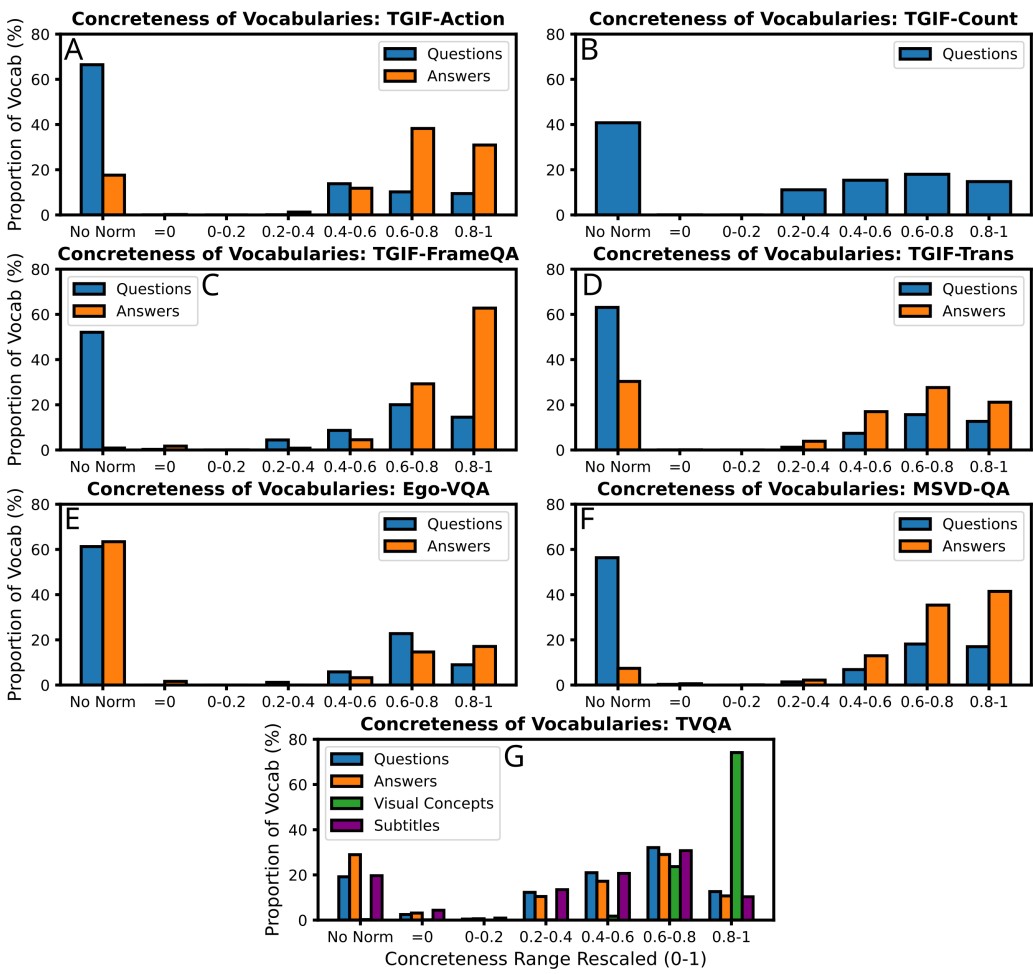

**Figure 12** **The relative abundance of the psycholinguistic 'concreteness' score in the vocabularies of each source of text in the video-QA datasets we experiment with.** Stopwords are not included. Concreteness scores are taken from the following datasets: MT40k (*Brysbaert, Warriner & Kuperman, 2013*), USF (*Nelson, Mcevoy & Schreiber, 1998*), SimLex999 (*Hill, Reichart & Korhonen, 2015*), Clark-Paivio (*Clark & Paivio, 2004*), Toronto Word Pool (*Friendly et al., 1982*), Chinese Word Norm Corpus (*Yee, 2017*), MEGAHR-Crossling (*Ljubešić, Fišer & Peti-Stantić, 2018*), Glasgow Norms (*Scott et al., 2017*; *Reilly & Kean, 2007*), and (*Sianipar, Groenestijn & Dijkstra, 2016*). The scores for each word are abstract = 0 and most concrete = 1, and the result averaged if more than 1 dataset has the same word.

consider for multimodal fusion in video-QA, most notably that attention mechanisms should be prioritised over BLP in text-vision fusion. We qualitatively show the potential for neurologically-motivated multimodal approaches in video-QA by identifying the relative abundance of psycholinguistically 'concrete' words in the vocabularies for the text components of the 4 video-QA datasets we experiment with. We would like to emphasise the importance of related neurological theories in deep learning and encourage researchers to explore Dual Coding Theory and the Two-Stream model of vision.

## ACKNOWLEDGEMENTS

We would like to thank Stuart White and Liz White for their support.

### Funding

This work was supported by the European Regional Development Fund (ERDF), and Carbon AI. The funders had no role in study design, data collection and analysis, decision to publish, or preparation of the manuscript.

### Grant Disclosures

The following grant information was disclosed by the authors:
The European Regional Development Fund (ERDF).
Carbon AI.

### Competing Interests

Alistair McLean is employed at Carbon AI, Middlesbrough.

### Author Contributions

- Thomas Winterbottom conceived and designed the experiments, performed the experiments, analyzed the data, performed the computation work, prepared figures and/or tables, authored or reviewed drafts of the paper, and approved the final draft.
- Sarah Xiao analyzed the data, authored or reviewed drafts of the paper, provided and reviewed the relevant neurological literature to this paper, and approved the final draft.
- Alistair McLean analyzed the data, authored or reviewed drafts of the paper, and approved the final draft.
- Noura Al Moubayed conceived and designed the experiments, analyzed the data, authored or reviewed drafts of the paper, and approved the final draft.

### Data Availability

All the code used to run all experiments and replicate all findings described in the article, including links and instructions of how to download each dataset from their respective owner's repositories, are available at GitHub: https://github.com/Jumperkables/trying_blp.

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
