# Peer review of "Bilinear pooling in video-QA: empirical challenges and motivational drift from neurological parallels"

_PeerJ Computer Science, doi:10.7717/peerj-cs.974_

## Round 0.1 · original submission · Major Revisions

We have received two consistent reports regarding the paper. A major revision will be needed before further processing. The revised paper will be sent to the reviewers for a further evaluation.

Reviewer 1 ·

Basic reporting

1. The expression of the article is clear and coherent. However, there are some typos. In line 106, 'abstracted' to 'abstract'. In line 423, 'eachother' to 'each other'. Some terms are not easy to understand, such as in line 455, 'ImageNet-style feature vectors'. Many experiment results in the tables are not rigorous, i.e., the baseline offsets in Tables 3 and 5 have no percentage sign.
2. The overall article is well organized. However, the structure in the discussion section is chaotic. The analysis of experimental results may be more reasonable in the experiment section, for readers to contact contextual content more easily.

Experimental design

1. The experimental results are relatively single, only quantitative analysis results are shown. There are no more qualitative or analytical experimental results.

Validity of the findings

1. All experimental results showed no improvement from baselines, which was a little bit strange. I doubt the correctness of the experimental setup, for the BLP has its rationality as the author proposed and has obvious performance improvement in image QA tasks[1].
[1] Ben-younes, H., Cade`ne, R., Cord, M., and Thome, N. (2017). Mutan: Multimodal tucker fusion for visual question answering. 2017 IEEE International Conference on Computer Vision (ICCV), pages 655 2631–2639.
2. The insights given in the article are relatively basic, and the discussion is not deep enough. The author did not give an overall summary or analysis of the multiple BLP methods. The author gives a brief introduction of the neurological parallels, however, their theoretical inspiration is not clearly explained. In addition, the reasons why the BLP does harm on the video QA task didn't convince me very well. If it does do harm, what kind of interaction method should we use in the future for cross-modality representation need to be discussed further.

Additional comments

no comment

·

Basic reporting

The paper focuses on the bilinear pooling problem in the context of Video-QA. The problem is indeed topical and subject to many research works. The topic is therefore appropriate and also important. The quality of the English language is not a problem and the paper is clearly written from both a literary and scientific point of view.
The structure of the manuscript is quite in accordance with the standard required by the journal and the figures are of good quality with good definition and do not suffer from any particular problem.
The datasets used are among the most appropriate and the availability of two of them is well indicated in the paper (TVQA and HME-VideoQA) but not for the others. There is no indication on the availability of the dual stream model proposed by the authors.
The title of the paper however raises an interesting and important question but the content of the paper does not provide any explanation or clear answer to the question induced by the title. This is one of the weaknesses of this paper that should be improved in order to have a real added value and a significant contribution of the work.

Experimental design

The subject matter is well within the scope of the paper and raises an important issue that is expressed in the title, namely "The limitations of bilinear pooling in video-QA". These limitations are indeed shown by means of experiments and comparisons with state-of-the-art methods. Unfortunately, there is no clear explanation of the underlying causes of this weakness of bilinear pooling except for a problem related to datasets, knowing that some of these datasets, such as MovieQA and YouTube2TextQA, seem, according to the literature, to provide the necessary for this kind of studies.
In my humble opinion, it would be necessary to explain the current failures of bilinear pooling which may be due to a problem of alignment of textual and video stream modalities with also temporal alignment or to the expression of queries. A conclusion on this subject would also be welcome and there are probably two among which one should decide. The first conclusion is that the bilinear pooling is not adapted to the problem by explaining why, but this conclusion, in my opinion is not the right one because it is enough to have the adequate datasets to solve part of the problem. The second conclusion should propose corrections to the existing models. The ideas proposed in the paragraph "PROPOSED AREAS OF RESEARCH" (line 607) remain mostly general and vague intuitions. More concrete and forceful proposals would give more weight to the paper.
The idea of demonstrating the weaknesses of a model is a very good approach in research, but again, these weaknesses should be clearly expressed, explained and alternatives or corrections proposed.

Validity of the findings

The scientific findings are at this stage quite light and need to be improved. The preceding comments may guide the authors in improving their contributions. The authors claim that, to their knowledge, they are the first to describe how important neurological theories, namely the dual coding theory and the two-stream vision model, are related to the history (and background) of modern multimodal deep learning practices. The thinking is certainly good but needs more formal framework and mathematical expression to be effectively exploited.

My suggestions to authors
1. Try to explain clearly where the current weaknesses of bilinear pooling lie and not just the dataset problem.
2. In the light of these weaknesses propose ideas for improvement or correction.
3. Formalize their idea of the dual coding theory and the dual stream vision model so that it can have a significant contribution in this area.

---

## Round 0.2 · accepted · Accept

The reviewers' comments have been carefully addressed. I recommend it for publication.

#